# The deca-GX$_3$ proteins Yae1-Lto1 function as adaptors recruiting the ABC protein Rli1 for iron-sulfur cluster insertion

Viktoria Désirée Paul[1], Ulrich Mühlenhoff[1], Martin Stümpfig[1], Jan Seebacher[2†], Karl G Kugler[2], Christian Renicke[3], Christof Taxis[3], Anne-Claude Gavin[2], Antonio J Pierik[1,4], Roland Lill[1,5]*

[1]Institut für Zytobiologie und Zytopathologie, Philipps-Universität, Marburg, Germany; [2]Structural and Computational Biology Unit, European Molecular Biology Laboratory, Heidelberg, Germany; [3]Fachbereich Biologie/Genetik, Philipps-Universität Marburg, Marburg, Germany; [4]Fachbereich Chemie, Technische Universität Kaiserslautern, Kaiserslautern, Germany; [5]LOEWE Zentrum für Synthetische Mikrobiologie SynMikro, Marburg, Germany

**Abstract** Cytosolic and nuclear iron-sulfur (Fe-S) proteins are involved in many essential pathways including translation and DNA maintenance. Their maturation requires the cytosolic Fe-S protein assembly (CIA) machinery. To identify new CIA proteins we employed systematic protein interaction approaches and discovered the essential proteins Yae1 and Lto1 as binding partners of the CIA targeting complex. Depletion of Yae1 or Lto1 results in defective Fe-S maturation of the ribosome-associated ABC protein Rli1, but surprisingly no other tested targets. Yae1 and Lto1 facilitate Fe-S cluster assembly on Rli1 in a chain of binding events. Lto1 uses its conserved C-terminal tryptophan for binding the CIA targeting complex, the deca-GX$_3$ motifs in both Yae1 and Lto1 facilitate their complex formation, and Yae1 recruits Rli1. Human YAE1D1 and the cancer-related ORAOV1 can replace their yeast counterparts demonstrating evolutionary conservation. Collectively, the Yae1-Lto1 complex functions as a target-specific adaptor that recruits apo-Rli1 to the generic CIA machinery.

*For correspondence: lill@staff.uni-marburg.de

Present address: †Friedrich Miescher Institute for Biomedical Research, Basel, Switzerland

Competing interests: The authors declare that no competing interests exist.

## Introduction

Maturation of iron-sulfur (Fe-S) proteins in (non-plant) eukaryotes depends on the coordinated action of complex proteinaceous machineries in mitochondria and cytosol (*Lill, 2009*; *Stehling and Lill, 2013*; *Lukes and Basu, 2015*; *Maio and Rouault, 2015*). The mitochondrial iron-sulfur cluster (ISC) assembly machinery not only generates the organellar Fe-S proteins but is also essential for the biogenesis of cytosolic and nuclear Fe-S proteins (*Kispal et al., 1999*; *Gerber et al., 2004*; *Biederbick et al., 2006*). The core part of the mitochondrial ISC machinery generates a sulfur-containing compound that is exported to the cytosol via the ABC transporter Atm1 (*Kispal et al., 1999*; *Lill et al., 2014*). Maturation of cytosolic and nuclear Fe-S proteins is further assisted by the cytosolic Fe-S protein assembly (CIA) machinery that is conserved in virtually all eukaryotes (*Sharma et al., 2010*; *Netz et al., 2014*; *Paul and Lill, 2015*).

Cell biological and biochemical investigations have revealed nine CIA proteins, and an initial mechanistic model of their function has been generated mainly from work in the model organism *Saccharomyces cerevisiae* and in human cell culture. The first known step of this biosynthetic process involves the de novo assembly of a [4Fe-4S] cluster on the scaffold protein complex consisting of the two related P-loop NTPases Cfd1 and Nbp35 (*Roy et al., 2003*; *Hausmann et al., 2005*; *Netz et al.,*

**eLife digest** Many proteins depend on small molecules called cofactors to be able to perform their roles in cells. One class of proteins—the iron-sulfur proteins—contain cofactors that are made of clusters of iron and sulfide ions. In yeast, humans and other eukaryotes, the clusters are assembled and incorporated into their target proteins by a group of assembly factors called the CIA machinery.

Several components of the CIA machinery have previously been identified and most of them appear to be core components that are needed to assemble many different proteins in cells. Since these iron-sulfur proteins are involved in important processes such as the production of proteins and the maintenance of DNA, losing of any of these CIA proteins tends to be lethal to the organism.

Paul et al. used several 'proteomic' techniques to study the assembly of iron-sulfur proteins in yeast and identified two new proteins called Yae1 and Lto1 that are involved in this process. Unlike other CIA proteins, Yae1 and Lto1 are only required for the assembly of just one particular iron-sulfur protein called Rli1, which is essential for the production of proteins. Most newly made iron-sulfur proteins can bind directly to a group of CIA proteins called the CIA targeting complex, but Rli1 cannot. The experiments show that Lto1 binds to both the CIA targeting complex and to Yae1, which in turn recruits the Rli1 to the CIA complex.

Paul et al. also show that humans have proteins that are very similar to Yae1 and Lto1. Inserting the human counterparts of Yae1 and Lto1 into yeast lacking these proteins could fully restore the assembly of iron-sulfur clusters into Rli1. This suggests that Yae1 and Lto1 proteins evolved in the common ancestors of fungi and humans and have changed little since. Taken together, Paul et al.'s findings reveal that Yae1 and Lto1 act as adaptors that link the rest of the CIA machinery to their specific target protein Rli1 in yeast and humans. A future challenge is to find out the three-dimensional structures of Yae1 and Lto1 to better understand how these proteins work and interact.

2007, 2012a; Basu et al., 2014). Cluster assembly additionally depends on the electron transfer chain composed of NADPH, the diflavin reductase Tah18 and the Fe-S protein Dre2 (Zhang et al., 2008; Netz et al., 2010). The second step includes the release of the [4Fe-4S] cluster from the scaffold complex, cluster transfer and subsequent insertion into apoproteins. These reactions are assisted by the iron-only hydrogenase-like protein Nar1 (human IOP1) which fulfils an intermediary, so far not well-defined function, by binding to both Nbp35 and the CIA targeting complex (Balk et al., 2004; Song and Lee, 2008). The latter is composed of the WD40-repeat protein Cia1, the DUF59 domain protein Cia2 and the HEAT-repeat protein Mms19. All three proteins are conserved from yeast to man, and physically associate with multiple Fe-S target proteins suggesting a direct function in cluster transfer and/or insertion (Balk et al., 2005; Weerapana et al., 2010; Gari et al., 2012; Stehling et al., 2012, 2013).

Impairment of either the ISC or CIA components leads to a maturation defect of numerous Fe-S proteins with a function in genome maintenance including DNA polymerases and DNA helicases. In turn, ISC or CIA depletion causes severe defects in DNA metabolism including DNA synthesis and repair, chromosome segregation and telomere length regulation (Veatch et al., 2009; Thierbach et al., 2010; Gari et al., 2012; Stehling et al., 2012). As a result, the DNA damage response pathway is activated. Another interesting target of the CIA machinery is the essential ABC protein Rli1 (RNase L inhibitor; human ABCE1). It belongs to the most conserved proteins in Eukarya and Archaea (Barthelme et al., 2007; Becker et al., 2012), and harbors an N-terminal ferredoxin-like domain with eight conserved cysteine residues that coordinate two [4Fe-4S] clusters (Karcher et al., 2005; Kispal et al., 2005; Barthelme et al., 2007). Depletion of Rli1 or disruption of Fe-S cofactor binding leads to nuclear export defects of both ribosomal subunits and consequently to translational arrest (Kispal et al., 2005; Yarunin et al., 2005). Moreover, Rli1 directly influences protein synthesis via different roles in translation initiation (Dong et al., 2004; Chen et al., 2006), termination (Khoshnevis et al., 2010; Shoemaker and Green, 2011), and splitting of ribosomal subunits (Barthelme et al., 2011; Shoemaker and Green, 2011; Becker et al., 2012). Maintenance of Rli1 function was recently suggested to be the 'Achilles' heel' of aerobic organisms, because Fe-S cofactor supply to Rli1 is sensitive to diverse pro-oxidants (Alhebshi et al., 2012).

We hypothesized that additional CIA factors may exist. These may include target-specific biogenesis factors that, similar to the mitochondrial ISC proteins Ind1, Nfu1, and BOLA3

(*Stehling and Lill, 2013*), assist the maturation of selected apoproteins. Such dedicated CIA proteins are expected to bind to both the late-acting CIA components and specific target Fe-S proteins. Here, we performed systematic protein interaction screens in *S. cerevisiae* to identify new CIA interaction partners. We describe the essential proteins Yae1 and Lto1 as binding partners of the CIA targeting complex. Previously, both proteins were shown to be associated with each other and to interact with the ribosome-associated Fe-S protein Rli1 (*Krogan et al., 2006*; *Zhai et al., 2014*). While the role of Yae1 is unknown, Lto1 ('required for biogenesis of the **l**arge ribosomal subunit and initiation of **t**ranslation in **o**xygen') was linked to the maturation of the 60S ribosomal subunit and translation initiation under aerobic conditions, yet its precise task remained unclear (*Zhai et al., 2014*). In this work, we functionally characterized the yeast and human Yae1-Lto1 complexes as novel target-specific CIA maturation factors for Fe-S cluster assembly on Rli1. Newly developed biochemical approaches and mutagenesis of important structural domains of these proteins elucidated a unique molecular mechanism of how these CIA proteins assist the Fe-S maturation process.

## Results

### Yae1 and Lto1 interact with the CIA targeting complex, in particular if Fe-S protein biogenesis is impaired

To systematically investigate the interaction network of the CIA proteins and to identify new interaction partners of the CIA machinery, we performed tandem affinity purifications (TAPs) in yeast (*Rigaut et al., 1999*; *Puig et al., 2001*) using the known CIA factors with a C-terminal TAP-tag as bait. Co-purified proteins were analyzed by mass spectrometry (*Figure 1—figure supplement 1A*). The procedure yielded numerous interactions between the various CIA components (*Figure 1—figure supplement 1B,C*). For instance, complex formation between Tah18 and Dre2 as well as between Cfd1 and Nbp35 were detected. Moreover, tight interactions were observed within the CIA targeting complex (Cia1-Cia2-Mms19). In keeping with functional observations (*Balk et al., 2005*; *Hausmann et al., 2005*), Nar1 was found to undergo interactions with both the early- and late-acting CIA components. Overall, this interaction pattern fully confirms previous findings on the CIA interactome, thus verifying our experimental strategy (*Netz et al., 2014*). Three additional protein interactions caught our attention. First, all three components of the CIA targeting complex showed an interaction with the Fe-S protein Rli1, a known target of the CIA machinery (*Figure 1—figure supplement 1B,C*) (*Balk et al., 2005*; *Gari et al., 2012*; *Stehling et al., 2012*). Second, by using several interaction analysis criteria these CIA factors were also found to associate with the essential proteins Yae1 and Lto1. Previous work had provided evidence that Yae1 and Lto1 interact with each other and form a complex with the essential ribosome-associated Fe-S protein Rli1 (*Krogan et al., 2006*; *Zhai et al., 2014*).

To verify the CIA protein interactions with Yae1 and Lto1, we expressed Myc- and HA-tagged versions of Yae1 and Lto1 in wild-type (WT) yeast for affinity purifications. Isolated proteins and their associated partners were analyzed by immunostaining. In addition to the tight interaction between Yae1 and Lto1 (*Figure 1A*, lanes 1 and 2), complex formation of these two proteins with any of the CIA targeting complex partners was observed when Lto1 and Yae1 were co-expressed. Affinity purification of the Yae1-Lto1 complex with the late-acting CIA proteins was significantly enhanced, when the analysis was performed in cells depleted of the early-acting CIA factor Nbp35 (*Figure 1A*, lanes 6–9). This indicated that the interactions of Yae1-Lto1 with CIA components are strengthened, when cytosolic Fe-S protein maturation is impaired. Nevertheless, the efficiency of Yae1-Lto1-Rli1 complex formation (*Zhai et al., 2014*) was not enhanced by Nbp35 depletion (*Figure 1B*), demonstrating that inactivation of the CIA activity affects the association of Yae1-Lto1 with the CIA targeting complex more efficiently than with Rli1. These findings verify our systematic interaction analyses and raise the question whether Yae1 and Lto1 are Fe-S protein targets or perform a function in cytosolic-nuclear Fe-S protein biogenesis.

### Yae1 and Lto1 specifically mature the Fe-S protein Rli1

Yae1 and Lto1 are conserved from yeast to man and show weak sequence similarity to each other in a central region of the protein (*Figure 2—figure supplement 1*). The proteins do not contain any conserved cysteine residues refuting the idea that they might function as recipient Fe-S proteins. To study the potential role of Yae1 and Lto1 in the maturation of cytosolic and nuclear Fe-S proteins, we employed an established in vivo $^{55}$Fe radiolabeling assay (*Kispal et al., 1999*). To this end, we

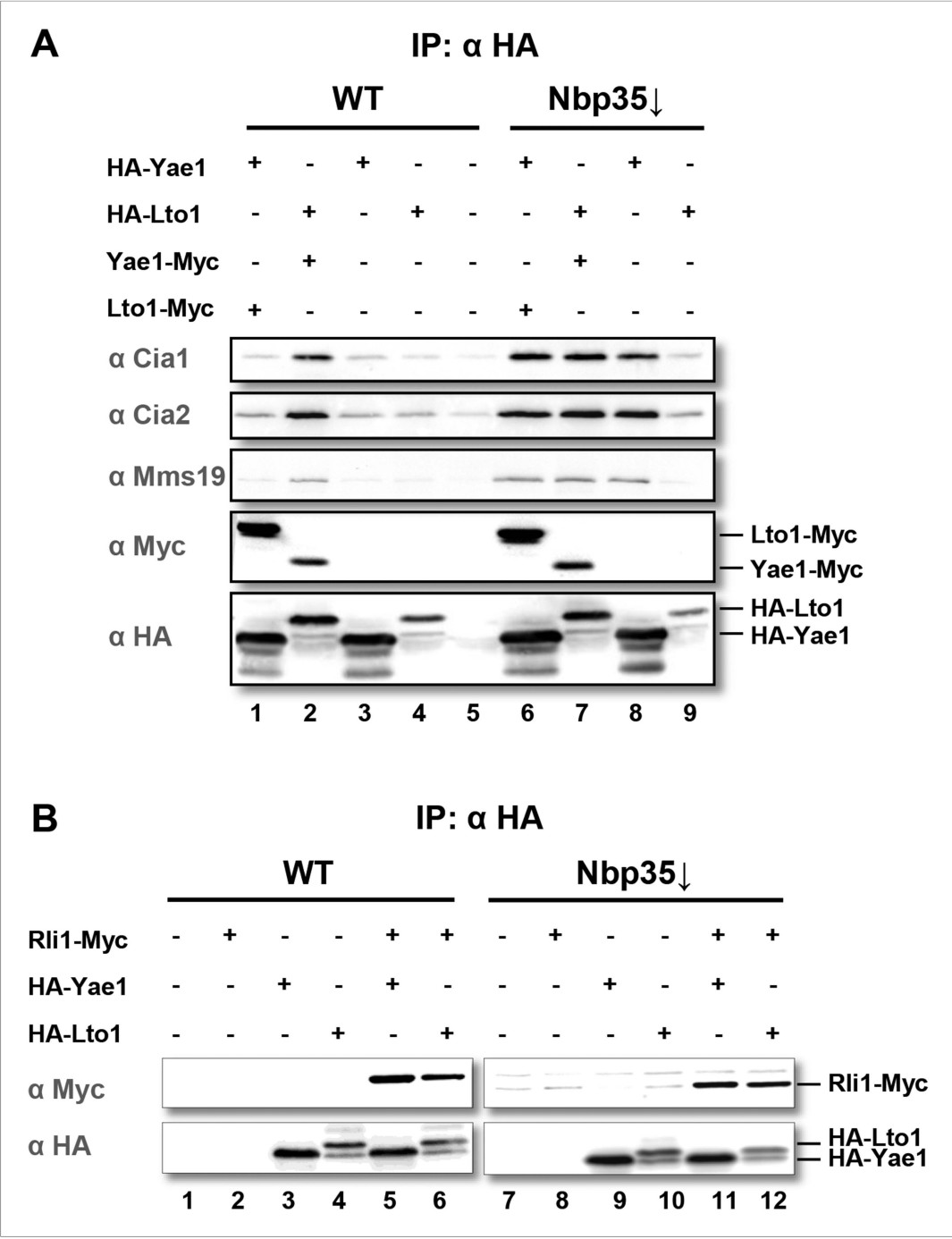

**Figure 1**. Yae1 and Lto1 interact with the cytosolic Fe-S protein assembly (CIA) targeting complex and the Fe-S protein Rli1. (**A**) Wild-type (WT) and Gal-NBP35 yeast cells were co-transformed with 2μ vectors containing either no insert or genes encoding HA-Yae1, HA-Lto1, Yae1-Myc and Lto1-Myc as indicated. Cells were cultivated in minimal medium containing glucose leading to Nbp35 depletion (↓) in Gal-NBP35 cells. Cell lysates were prepared, and a HA-tag immunoprecipitation (IP) was performed. The immunoprecipitate was analyzed for the indicated proteins or tags by immunoblotting. (**B**) WT and Gal-NBP35 yeast cells were co-transformed with 2μ vectors containing either no insert or genes encoding HA-Yae1, HA-Lto1 and Rli1-Myc and treated as in part **A**. The lower band in HA-Lto1-containing lanes is a HA-Lto1 degradation product.

The following figure supplement is available for figure 1:

**Figure supplement 1**. Yae1 and Lto1 interact with the CIA targeting complex and with the Fe-S protein Rli1.

constructed regulatable *GALL*-promoter-containing yeast strains (Gal-YAE1, Gal-LTO1) which allowed expression of the proteins by cell growth in the presence of galactose and their depletion by growth in the presence of glucose (*Janke et al., 2004*). In this context, we noted that the start codon of the *LTO1* gene was erroneously annotated (for further details of this aspect see *Figure 2—figure supplements 2, 3*, and 'Materials and methods'). After [55]Fe radiolabeling of WT, Gal-YAE1, Gal-LTO1 and control Gal-NBP35 cells, selected cytosolic and nuclear Fe-S proteins were immunoprecipitated. The protein-associated radioactivity was determined by scintillation counting as a measure of Fe-S cluster assembly. Depletion of Yae1 and Lto1 led to a strong diminution of [55]Fe association with Rli1, and this decrease was comparable to that observed upon depletion of Nbp35 (*Figure 2A*). Despite structural similarities between Yae1 and Lto1, these proteins could not mutually replace each other in Rli1 maturation, even after overexpression (*Figure 2B*). Surprisingly, no significant decline in [55]Fe-S cluster formation was detectable for three canonical nuclear Fe-S target proteins (Rad3, Ntg2, and Pol3) upon Yae1 or Lto1 depletion (*Figure 2C–E*). As expected, these proteins were not assembled with a [55]Fe-S cluster in cells depleted for the CIA component Nbp35. Immunostaining showed that the levels of Rli1 and the other Fe-S proteins remained unchanged upon depletion of Yae1, Lto1 and Nbp35, with the notable exception of Pol3 in Nbp35-depleted cells due to the known instability of its apoform (*Figure 2F*) (*Netz et al., 2012b*). In conclusion, our in vivo [55]Fe labeling analysis suggests that Yae1 and Lto1 are specifically required for Fe-S cluster association with Rli1, yet these factors are dispensable for assembly of other Fe-S proteins.

To further investigate the intriguing target specificity of Yae1 and Lto1 in cytosolic-nuclear Fe-S protein assembly, we analyzed the effects of Yae1 or Lto1 deficiency on the activities of the cytosolic Fe-S enzymes isopropylmalate isomerase Leu1 and sulfite reductase. While depletion of Nbp35 strongly diminished the Leu1 activity, depletion of Yae1 or Lto1 had hardly any effect (*Figure 3A*). Likewise, Yae1 and Lto1 were not required for sulfite reductase activity as estimated by an in vivo color assay (*Figure 3B*). Further, we analyzed Fe-S cluster assembly on the early-acting CIA proteins Nbp35 and Nar1 in order to explore whether Yae1 or Lto1 deficiency affects the function of the CIA system (*Figure 3C–E*). As estimated by [55]Fe radiolabeling, neither Yae1 nor Lto1 were involved in the maturation of these two Fe-S cluster-containing CIA components. In contrast, depletion of the mitochondrial ISC component Yah1 strongly diminished Fe-S cluster assembly on Nbp35 and Nar1. These findings indicate that Yae1-Lto1 operate downstream of these CIA factors in Rli1 Fe-S cluster association. Finally, activities of the non-Fe-S protein alcohol dehydrogenase and the mitochondrial Fe-S protein aconitase were unchanged upon depletion of both Yae1 and Lto1 (*Figure 3—figure supplement 1*). Taken together, these results, combined with the tight association of Yae1 and Lto1 to the CIA targeting complex, suggest that both proteins are specific for Fe-S cluster formation on Rli1, and thus act late in the CIA pathway. A similar Fe-S protein target specificity is unprecedented in yeast.

Rli1 and its essential Fe-S cofactors are involved in the export of ribosomal subunits from the nucleus and are required for various aspects of protein translation (*Hopfner, 2012*). To address the influence of Yae1 and Lto1 on cytosolic protein synthesis, yeast cells containing or lacking Yae1 and Lto1 were briefly radiolabelled with [35]S-methionine. Cell extracts were prepared and analyzed for incorporation of the radiolabeled amino acid into nascent proteins. Depletion of Yae1 or Lto1 impaired protein translation in a similar fashion as the deficiency in Rli1 (*Figure 3F*; cf. [*Kispal et al., 2005*]). We conclude that the influence of Yae1-Lto1 on protein biosynthesis is explained by their role in Fe-S cluster formation on Rli1.

## Yae1 is a Fe-S cluster maturation rather than stabilization factor

The specific requirement of Yae1 and Lto1 for the presence of Fe-S clusters on Rli1 suggested that these either act as specific CIA maturation factors or stabilize the assembled Fe-S clusters on Rli1. Interestingly, assembly of Rli1 was previously shown to be sensitive to oxidative stress-inducing agents such as paraquat, which may destroy the ROS-labile Fe-S cofactors of Rli1 (*Alhebshi et al., 2012*). Further, Yae1 and Lto1 were found to be essential under aerobic but not under anaerobic growth conditions (*Snoek and Steensma, 2006*). These findings may indicate a function of these proteins in oxidative stress protection of the Rli1 Fe-S clusters.

To discriminate between a maturation or stabilization function of Yae1 or Lto1 for Rli1's Fe-S clusters, we established a method which allowed us to rapidly degrade Yae1 or Lto1 and then follow the fate of Rli1's [55]Fe-S clusters. Targeted degradation of Yae1 and Lto1 was achieved by fusing

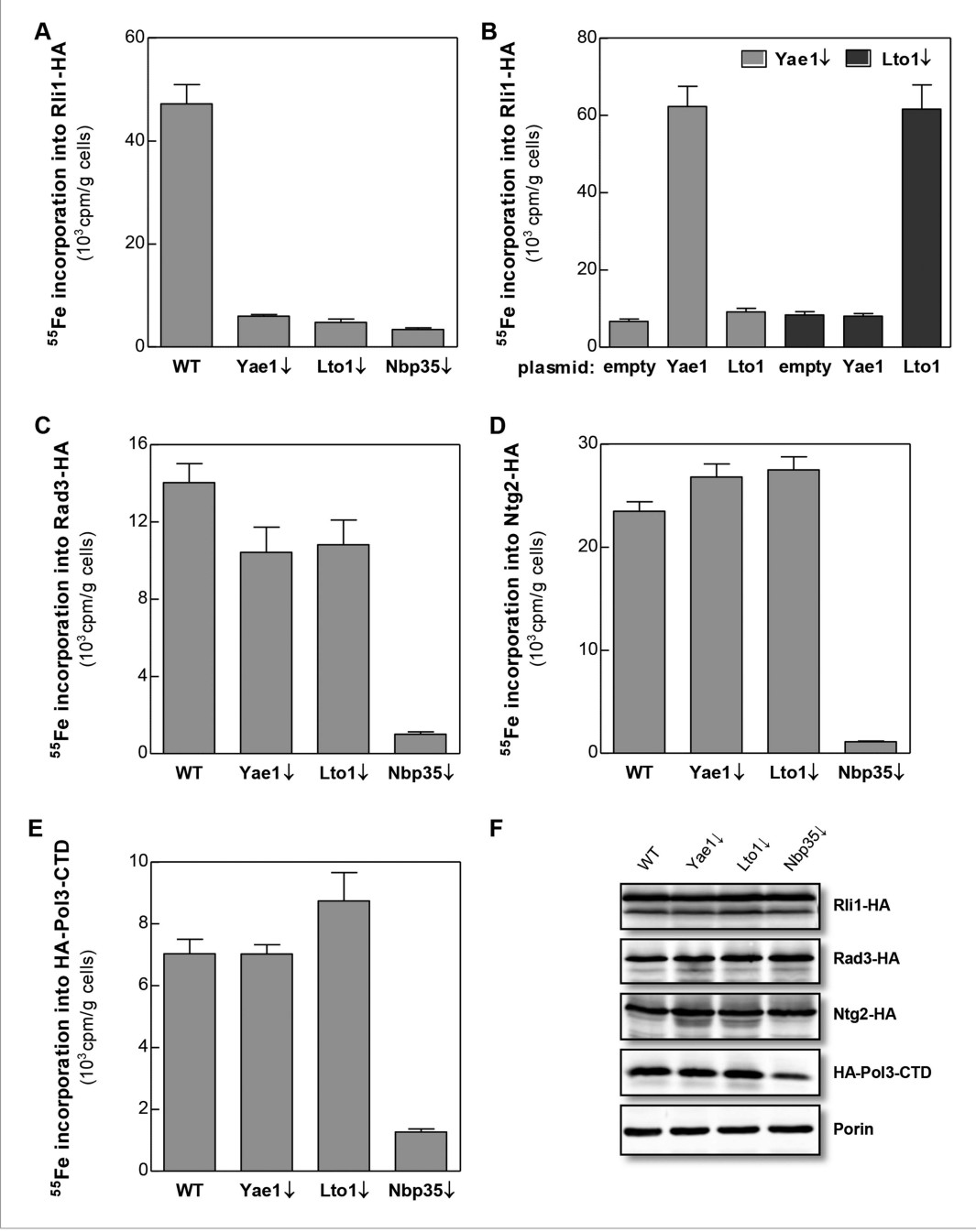

**Figure 2.** Yae1 and Lto1 specifically mediate Fe-S cluster association with Rli1. WT, Gal-YAE1, Gal-LTO1 and Gal-NBP35 yeast cells were transformed with plasmids encoding Rli1-HA (**A**, **B**), Rad3-HA (**C**), Ntg2-HA (**D**) and the C-terminal domain of Pol3 N-terminally fused to HA (**E**). To increase the amount of Pol3-CTD-bound Fe-S clusters the accessory subunit Pol31 was co-expressed (**Netz et al., 2012b**). In part **B** Gal-YAE1 and Gal-LTO1 cells contained 2µ plasmid-borne *YAE1* or *LTO1* or the empty vector. Cells were cultivated for 24 hr in glucose-containing minimal medium for depletion of the respective proteins (↓). After an additional 16 hr in minimal medium lacking iron cells were radiolabeled for 2 hr with $^{55}FeCl_3$, cell extracts were prepared, and the Fe-S target proteins were immunoprecipitated with anti-HA antibodies. The amount of $^{55}Fe$ associated with target proteins was quantified by scintillation counting. Error bars indicate the SEM (n > 4). (**F**) A representative immunostain of the indicated proteins from parts **A**, **C**–**E**.

The following figure supplements are available for figure 2:

*Figure 2. continued on next page*

*Figure 2. Continued*

**Figure supplement 1**. Yae1 and Lto1 contain a deca-GX$_3$ sequence motif that is conserved in eukaryotes.

**Figure supplement 2**. Generation of a regulatable strain for depletion of Lto1 questions the previously predicted physiological translation start site of *LTO1*.

**Figure supplement 3**. Determination of the correct physiological translation start site of *LTO1*.

a photosensitive degron (psd) to their C-termini. This degron is composed of the light-reactive LOV2 domain of *Arabidopsis thaliana* phot1 and the murine ornithine decarboxylase-like degradation sequence cODC1 (*Renicke et al., 2013*). Exposure to blue light triggers proteasomal degradation of the psd-fusion proteins (*Figure 4A*). Induced degradation of Yae1-psd and Lto1-psd was maximal after 2 hr of blue light exposure (*Figure 4—figure supplement 1A*). The extent of Yae1 and Lto1 depletion affected the growth rate of the mutant cells only weakly (*Figure 4—figure supplement 1B*). To further increase the depletion efficiency, cells were treated with the translation inhibitor cycloheximide during light exposure to fully inhibit synthesis of Yae1 protein (*Renicke et al., 2013*). This trick further decreased the amount of Yae1 (compare immunoblots of *Figure 4B* and *Figure 4—figure supplement 1A*). We then investigated the effect of light-induced Yae1 or Lto1 depletion on the de novo assembly of Rli1's Fe-S clusters by employing our $^{55}$Fe radiolabeling procedure. Light-induced depletion of Yae1 caused a strong reduction of $^{55}$Fe-S cluster insertion into Rli1 in comparison to the non-exposed sample (*Figure 4B*). In contrast, the amount of $^{55}$Fe associated with Leu1 was unchanged upon exposure to blue light. The latter result showed that cycloheximide had no general negative effect on $^{55}$Fe-S cluster insertion into apoproteins. The Lto1-psd mutant strain could not be used in this approach because Rli1 incorporated only little $^{55}$Fe suggesting that the C-terminal psd module interfered with normal function of Lto1. Collectively, these results demonstrate that the light-induced degradation of Yae1 leads to a similar diminution of Fe-S clusters on Rli1 as that observed for Gal-YAE1 cells (cf. *Figure 2A*).

We then investigated whether Yae1 might have a stabilizing effect on pre-assembled $^{55}$Fe-S clusters of Rli1. WT and Yae1-psd cells were radiolabeled with $^{55}$Fe in the dark, and then exposed to blue light to induce the rapid degradation of Yae1. A portion of the cells was harvested immediately after the radiolabeling reaction, and $^{55}$Fe associated with Rli1 and Leu1 was assessed (*Figure 4C*, t$_0$). The remaining cells were incubated for 4 hr in the absence or presence of blue light, and $^{55}$Fe association to Rli1 and Leu1 was determined (*Figure 4C*). Even though the amount of Rli1-bound $^{55}$Fe-S cluster dropped by 20% during the 4 hr incubation (see below), exposure to blue light had no significant detrimental effect on cluster stability. Similar results were observed for the control Fe-S protein Leu1, even though its Fe-S clusters appeared to be more stable over the 4 hr incubation period (*Figure 4C*). These results strongly suggest that Yae1 deficiency does not significantly affect the Fe-S cluster stability of Rli1. Hence, Yae1 functions as a specific CIA maturation factor for Rli1 rather than a Fe-S cluster-stabilizing protein.

## Why are *YAE1* and *LTO1* non-essential under anaerobic conditions?

Most CIA components are essential for cell viability (*Netz et al., 2014*). Strikingly, Yae1 and Lto1 are indispensable under aerobic, but not under anaerobic growth conditions (*Snoek and Steensma, 2006*) (*Figure 5A*). This raises the question whether Yae1 and Lto1 are dispensable for Fe-S cluster assembly on Rli1 under anaerobic conditions. When we analyzed the de novo $^{55}$Fe-S cluster assembly of Rli1 under anaerobic conditions, depletion of Yae1 or Lto1 led to similar Fe-S cluster assembly defects as depletion of Nbp35 (*Figure 5B*). These effects were not caused by diminished Rli1 protein levels in the different cell types (inset), demonstrating that Yae1 and Lto1 are involved in Rli1 maturation also under anaerobic conditions.

These observations may suggest that Rli1 is not essential under anaerobic conditions (*Zhai et al., 2014*). To test this idea, Rli1 was depleted under anaerobic conditions in the tetracycline-repressible Tet-RLI1 strain (*Kispal et al., 2005*). No growth was observed, unless cells were transformed with a plasmid containing WT *RLI1* (*Figure 5C*, top rows) demonstrating that Rli1 is essential even under

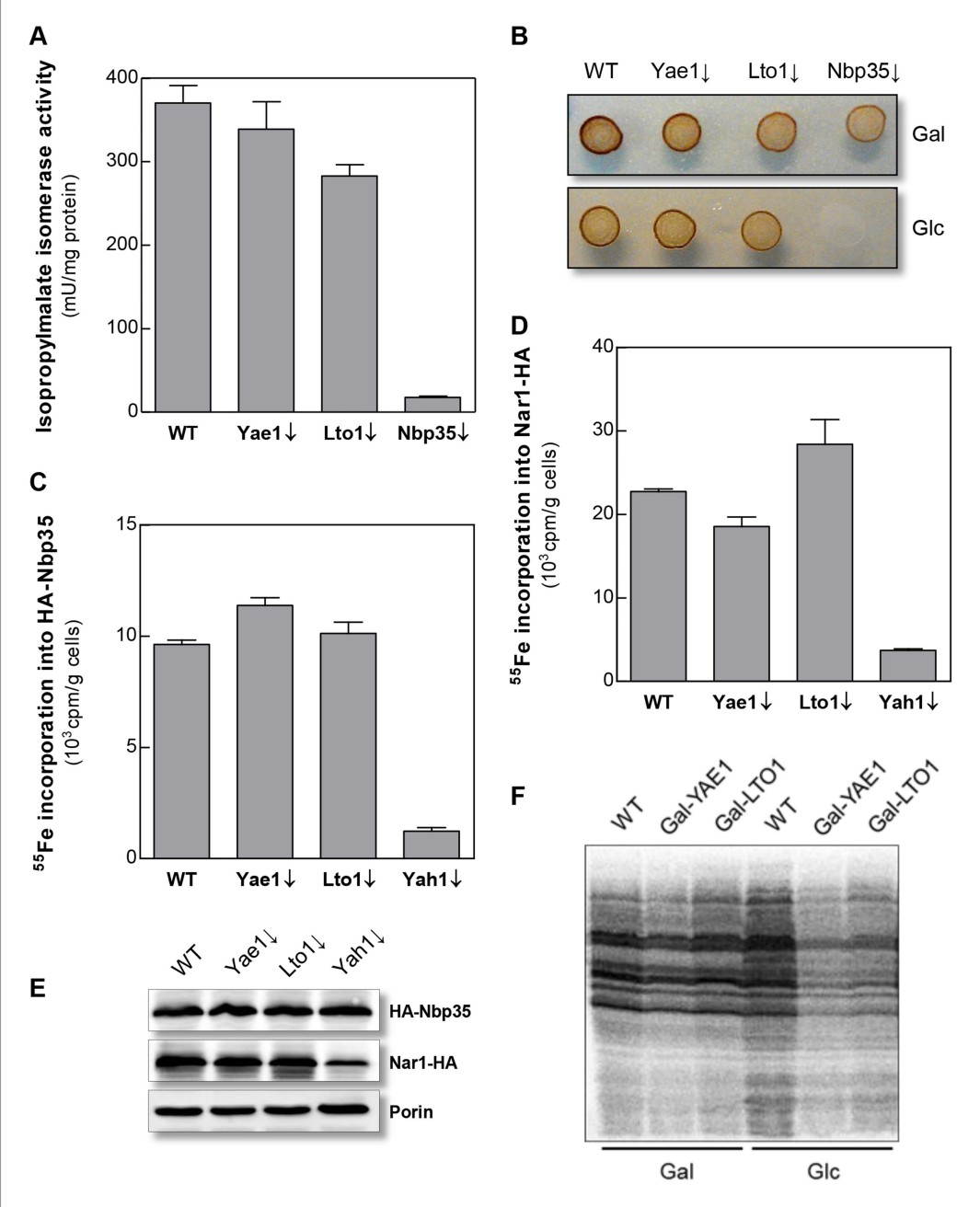

**Figure 3**. Yae1 and Lto1 do not perform a general role in Fe-S protein maturation. (**A**) WT, Gal-YAE1, Gal-LTO1 and Gal-NBP35 cells were cultivated on glucose-containing minimal medium for 40 hr. Cell extracts were analyzed for the activities of isopropylmalate isomerase (Leu1). (**B**) Cells were analyzed for sulfite reductase activity in vivo after growth for 3 days on minimal medium agar plates supplemented with ammonium bismuth citrate and sodium sulfite. Sulfide produced by sulfite reductase yields the brown precipitate $Bi_2S_3$. (**C**, **D**) WT, Gal-YAE1, Gal-LTO1 and Gal-YAH1 cells were transformed with plasmids containing genes encoding HA-Nbp35 (**C**) and Nar1-HA (**D**). Cells were radiolabeled with $^{55}FeCl_3$, and $^{55}$Fe-S protein formation was estimated as in *Figure 2*. (**E**) A representative immunostain of the indicated proteins from parts **C**, **D**. (**F**) WT, Gal-YAE1 and Gal-LTO1 cells were cultivated in rich medium containing galactose (Gal) or glucose (Glc) for 40 hr, and radiolabelled with $^{35}$S-methionine for 10 min at 30°C. Extracts were prepared, and the protein synthesis efficiency was analyzed by SDS-PAGE and autoradiography. Error bars indicate the SEM (n > 4).

The following figure supplement is available for figure 3:

*Figure 3. continued on next page*

Figure 3. Continued

**Figure supplement 1**. Depletion of Yae1, Lto1 and Nbp35 does not affect enzyme activities of the cytosolic alcohol dehydrogenase and the mitochondrial Fe-S enzyme aconitase.

anaerobiosis. A clue how to explain the non-essentiality of Yae1-Lto1 in the absence of oxygen came from the inspection of Rli1 mutant proteins in which cysteine residues 25 and 61 were exchanged either alone or simultaneously to serine (*Kispal et al., 2005*; *Zhai et al., 2014*). These residues coordinate two different [4Fe-4S] clusters of Rli1 (*Barthelme et al., 2007*). Depleted Tet-RLI1 cells expressing single cysteine mutants of *RLI1* were viable under anaerobic but not aerobic conditions (*Figure 5C*, middle rows). In contrast, the *RLI1* double mutation rendered cells inviable under both conditions (bottom row). These results show that single mutations of *RLI1* surprisingly support WT growth under anaerobic conditions (see also [*Zhai et al., 2014*] for the C25S mutation), apparently because Fe-S cluster binding to these Rli1 proteins is maintained despite the loss of one cluster coordination site. Such a behavior is not unusual for Fe-S proteins (*Urzica et al., 2009*; *Netz et al., 2012a*, *2012b*). Under aerobic conditions, and even more so in the presence of oxidants (*Alhebshi et al., 2012*), the Rli1 cluster binding stability is too low to sustain growth. The observed higher stability of the Rli1 Fe-S clusters in the absence of oxygen may readily explain the non-essentiality of Yae1 and Lto1, also indicating that their requirement in Fe-S cluster insertion into Rli1 can be bypassed to some extent under anaerobic conditions.

## Lto1 interacts with Yae1 via its unique deca-GX$_3$ motif and with the CIA targeting complex via its C-terminal tryptophan residue

We next investigated the functional importance of characteristic structural features of Yae1 and Lto1. Both proteins contain an evolutionary conserved deca-GX$_3$ motif of 40 residues (*Figure 6A*) that is not found in any other eukaryotic protein. Lto1 additionally carries a conserved C-terminal tryptophan (phenylalanine in some organisms). We generated Lto1 mutant proteins at positions G17;G21, G33; G37;G41 and G49;G53 of the deca-GX$_3$ motif, and exchanged the conserved aspartate D4 and the C-terminal tryptophan to alanine (*Figure 6A*). The HA-tagged Lto1 mutant versions were co-expressed together with Yae1-Myc to investigate their interaction. Since CIA protein depletion increased the complex formation between the CIA targeting complex and Yae1-Lto1 (cf. *Figure 1A*), we used Nbp35-depleted Gal-NBP35 cells for this analysis, yet similar results were observed with WT cells. Protein amounts of HA-Lto1 were not affected by these substitutions. Immunoprecipitation (IP) showed that Lto1 mutations at positions D4 or G17;G21 slightly affected the interaction with the CIA targeting complex in comparison to WT Lto1, whereas the mutations at positions G33;G37;G41 and G49;G53 almost fully abrogated this association. Likewise, these Lto1 mutations decreased complex formation with Yae1 (*Figure 6B*). Interestingly, mutation of the C-terminal tryptophan strongly impaired complex formation with Cia1, Cia2 and Mms19, but not with Yae1.

To study the consequences of these mutations for Lto1 function, we analyzed their influence on Rli1 maturation by $^{55}$Fe radiolabeling of Lto1-depleted cells producing the modified Lto1 versions (*Figure 6C*). Mutations of residues D4 or G17;G21 had only mild effects on $^{55}$Fe-S cluster assembly of Rli1. In contrast, mutations within the middle or C-terminal region of the deca-GX$_3$ motif or exchange of the C-terminal tryptophan severely impaired Rli1 maturation. Collectively, these data show that the deca-GX$_3$ domain is crucial for Lto1 complex formation with both Yae1 and the CIA targeting complex, while the C-terminal tryptophan of Lto1 is specifically required for the interaction with the CIA targeting complex. These impaired protein interactions account for the functional deficit of mutated Lto1 in Rli1 maturation, suggesting that the deca-GX$_3$ motif-tethered Lto1-Yae1 complex binds to the CIA targeting complex via the Lto1 tryptophan and to Rli1 via Yae1 (*Zhai et al., 2014*).

## The function of the Yae1-Lto1 complex is conserved from yeast to man

Searches of the human genome for homologs of yeast Yae1 and Lto1 identified YAE1D1 (Yae1 domain-containing protein 1) and ORAOV1 (oral cancer-overexpressed protein 1, [*Huang et al., 2002*; *Jiang et al., 2008*]), respectively (*Figure 2—figure supplement 1*). We previously have found these

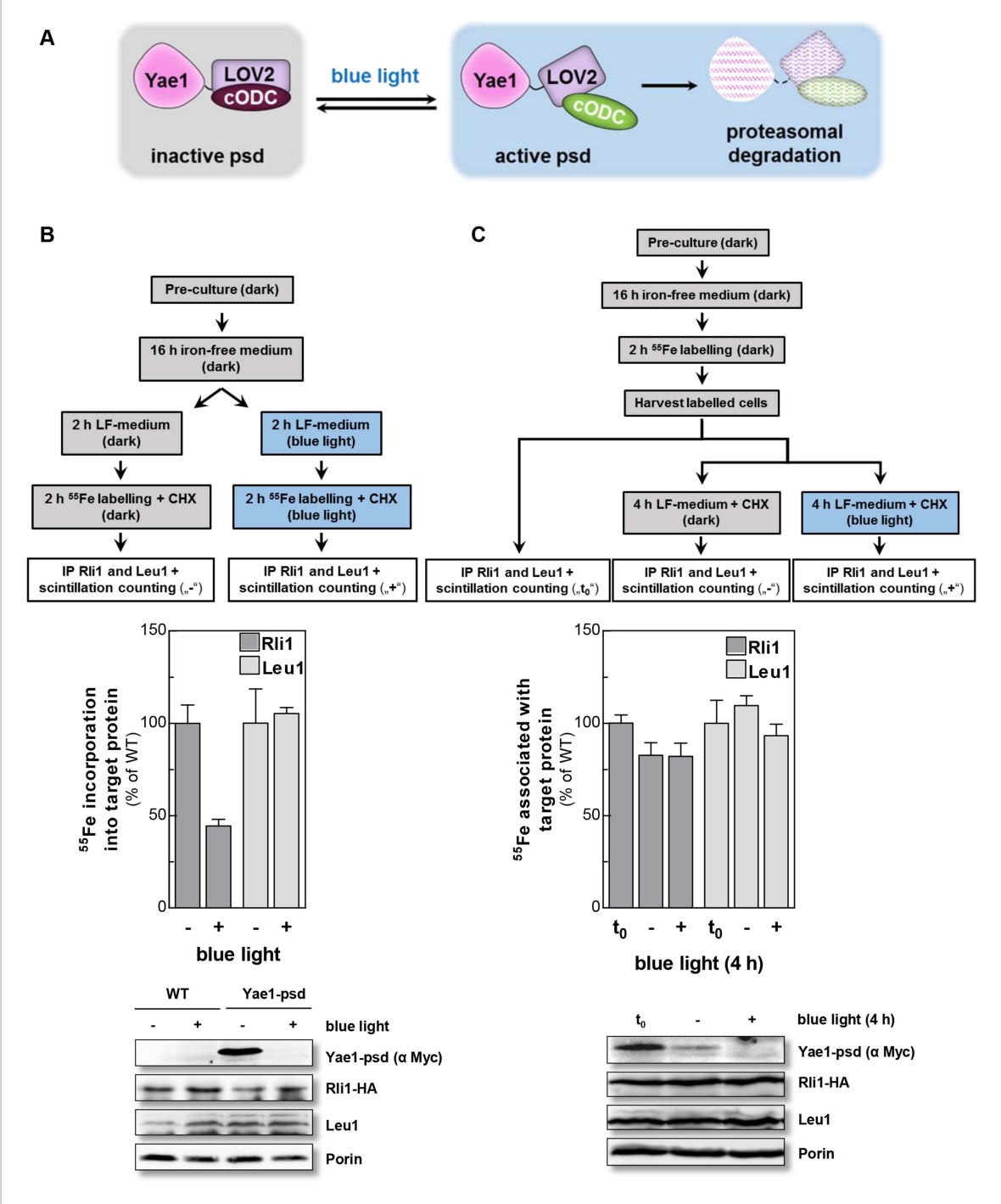

**Figure 4**. Yae1 is a Fe-S cluster maturation rather than stabilization factor for Rli1. (**A**) Schematic representation of the light-induced rapid degradation of the Yae1-3Myc-AtLOV2-cODC1 fusion protein (psd; photosensitive degron). In the dark the cODC1 degron is inactive and thus the fusion protein is stable. Irradiation with blue light induces a structural rearrangement within the LOV2 domain leading to the activation of the degron and direct degradation of the fusion protein by the 26S proteasome (adapted from [*Renicke et al., 2013*]). (**B**) WT and Yae1-psd (photosensitive degron) cells were transformed with a 2μ plasmid encoding *RLI1-HA*. Cells were grown overnight in iron-free minimal medium in the dark. Half of the cells (0.5 g) were exposed to blue light in low fluorescence medium (LFM), and the other half was kept in the dark for 2 hr. Cycloheximide (200 μg/ml) was added, and cells were radiolabeled with $^{55}$Fe with or without blue light irradiation. The amount of $^{55}$Fe associated with Rli1 or Leu1 was quantified by IP and scintillation counting (cf. *Figure 2*). The radioactivity associated with Fe-S proteins in Yae1-psd mutant cells is presented relative to that of WT cells. Error bars indicate the SEM (n > 3). The bottom part shows a representative immunostain of the indicated proteins in cell extracts. (**C**) Three independent cultures of WT and Yae1-psd cells were grown overnight in iron-free minimal medium in the dark and radiolabeled with $^{55}$Fe for 2 hr. Culture 1 was assessed immediately for $^{55}$Fe-S cluster

*Figure 4. continued on next page*

*Figure 4. Continued*

association to Rli1-HA or Leu1 by IP and scintillation counting ($t_0$). Cultures 2 and 3 were washed with $H_2O$ and supplemented in LFM with cycloheximide. Cells were kept in the dark (−) or exposed to blue light (+) for additional 4 hr, and then analyzed for $^{55}$Fe-S cluster association. The bottom part shows a representative immunostain of the indicated proteins in extracts of Yae1-psd cells.

The following figure supplement is available for figure 4:

**Figure supplement 1**. Fusion of Yae1 and Lto1 with a photosensitive degron allows their efficient degradation upon irradiation with blue light.

proteins amongst the components interacting with the human CIA targeting complex (*Stehling et al., 2012*, *2013*). We investigated whether the two human proteins can replace the essential function of their yeast counterparts upon expression in Gal-YAE1 or Gal-LTO1 cells. Neither YAE1D1 nor ORAOV1 restored the growth defects of yeast cells depleted for Yae1 or Lto1 (*Figure 7A*). In contrast, co-expression of both *YAE1D1* and *ORAOV1* fully restored growth, indicating the function of these proteins as a homologous complex. As expected from the lack of growth rescue, $^{55}$Fe-S cluster assembly on Rli1 was not improved upon expression of the single human proteins in Yae1- or Lto1-depleted yeast cells (*Figure 7B*). However, co-expression of both human complex partners restored Rli1 maturation to WT efficiency (*Figure 7C*). These data demonstrate that the human YAE1D1-ORAOV1 complex functionally replaces the yeast counterpart suggesting that the human proteins perform a conserved function in the maturation of the Rli1 homolog ABCE1.

## Discussion

In this work we have identified and functionally characterized the essential proteins Yae1 and Lto1 as two novel members of the CIA machinery. Their function is unique, because, unlike the previously identified CIA proteins, these components do not act as general Fe-S maturation factors. Rather, they function downstream of the known CIA proteins by specifically assembling the essential Fe-S protein Rli1. Our findings allowed us to expand the current model of CIA (*Figure 8*) (*Paul and Lill, 2014*). Yae1 and Lto1 form a complex that acts as a dedicated adaptor to recruit Rli1 to the late-acting part of the CIA machinery. This model was derived from the interaction of Yae1-Lto1 with both the CIA targeting complex (Cia1-Cia2-Mms19) and the Rli1 client. The high target specificity of Yae1-Lto1 is further documented by the fact that the complex is also dispensable for the maturation of Fe-S cluster-containing CIA proteins such as Nbp35 and Nar1. Yae1 and Lto1 perform non-overlapping, individual functions, since even upon overexpression they cannot mutually complement each other, despite some relationship in amino acid sequence (see below). Nevertheless, their function is conserved in higher eukaryotes as the human proteins YAE1D1 and ORAOV1 can substitute their yeast counterparts, but only when co-expressed, again documenting the function as a complex. Collectively, these data demonstrate that the Yae1-Lto1 complex acts as a conserved specificity factor for Fe-S cluster maturation of the highly conserved eukaryotic protein Rli1 (human ABCE1).

The amino acid sequences of Yae1 and Lto1 do not contain any characteristic similarities to other eukaryotic proteins, yet both proteins share the short deca-$GX_3$ sequence motif that is unique for these proteins in eukaryotes. A similar, but shorter '$GX_3G$' motif has been recognized in a number of membrane proteins, where the $GX_3G$ motif forms trans-membrane helix dimerization by inter-helical hydrogen bonds (*Senes et al., 2004*). The function of the deca-$GX_3$ motif has not been explored until now, but we speculate that it serves a role in Yae1-Lto1 heterodimer formation as in membrane proteins. The deca-$GX_3$ motif is also found in bacterial FliH, a protein involved in the assembly and substrate export of flagella (*Bai et al., 2014*). FliH forms a dimer and transiently binds to the ATPase FliI. Bacterial FliI is similar in structure to the αβ-subunits of $F_1F_0$ ATPase and hence not related to the ABC-type ATPase Rli1, leaving open any functional relevance of the deca-$GX_3$ protein—ATPase interaction. Our Lto1 mutational studies demonstrated that the deca-$GX_3$ motif is functionally important for Yae1-Lto1 complex formation, and consequently for complex interaction with the CIA machinery (*Figure 8*). Exchanges of conserved glycine residues of Lto1, for instance in the middle and C-terminal parts of the deca-$GX_3$ motif, strongly affected its association with both Yae1 and the CIA targeting complex, thus explaining the strongly impaired efficiency of these Lto1 variants to support Fe-S cluster assembly on Rli1. In contrast, mutation of two N-terminal glycine residues caused only

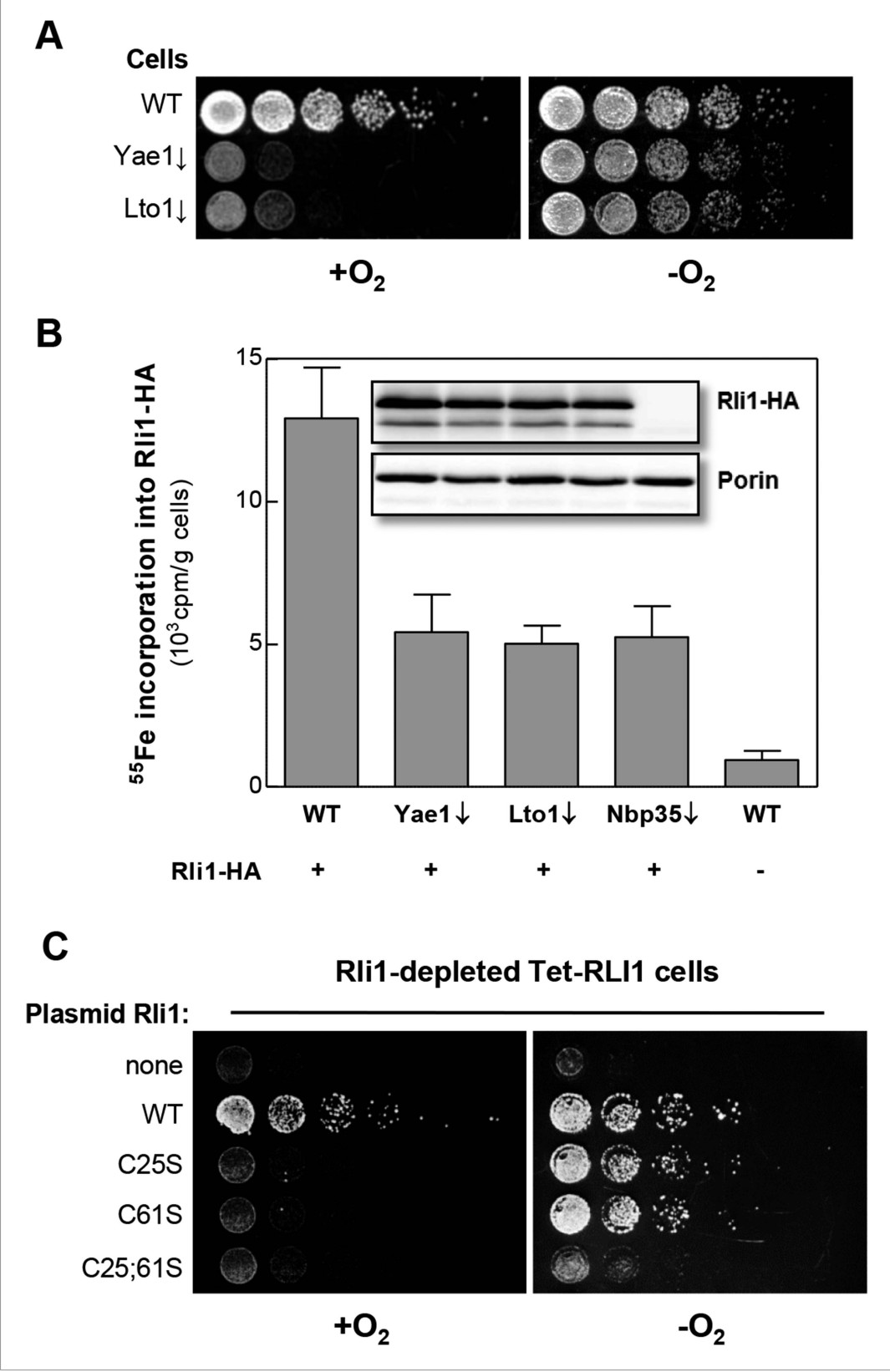

**Figure 5**. Yae1 and Lto1 are required for Fe-S cluster assembly on Rli1 also under anaerobic conditions. (**A**) The indicated cells were cultivated overnight in rich medium containing glucose. Serial dilutions (1:5) were spotted onto
*Figure 5. continued on next page*

*Figure 5. Continued*

glucose-containing rich medium agar plates. Growth was in the presence or absence of $O_2$. (**B**) WT, Gal-YAE1, Gal-LTO1 and Gal-NBP35 cells were transformed with plasmids encoding *RLI1-HA* (+) or no gene (−). Cells were cultivated for 30 hr in glucose-containing minimal medium under anaerobic conditions and transferred to minimal medium lacking iron for additional 16 hr. After radiolabeling for 2 hr with $^{55}Fe$ under anaerobic conditions, cell extracts were prepared anaerobically and analyzed for $^{55}Fe$ incorporation into Rli1-HA (cf. *Figure 2*). Error bars indicate the SEM (n > 3). The inset shows a representative immunostain of Rli1-HA and porin. (**C**) Tet-RLI1 cells were transformed with an empty vector, a plasmid containing *RLI1* under its native promoter (300 bp upstream of *RLI1*) or plasmids containing *RLI1* coding for cysteine to serine mutants of Rli1 as indicated. Cells were cultivated in glucose-containing minimal medium for 16 hr. Serial dilutions (1:5) were spotted under anaerobic conditions onto glucose-containing minimal medium agar plates supplemented with 5 µg/ml doxycycline to deplete endogenous Rli1. Further growth was in the presence or absence of $O_2$.

moderate effects on complex formation and Rli1 maturation suggesting some residual function of this disrupted deca-GX$_3$ motif.

A strikingly different result was obtained for mutation of the conserved C-terminal tryptophan residue of Lto1. Exchange to alanine had no detectable effect on complex formation with Yae1, yet the association of the Yae1-Lto1 complex with the CIA targeting complex was severely disrupted. The decreased Fe-S cluster assembly of Rli1 observed for this Lto1 W → A mutation is therefore best explained by the weakened contact of Lto1 to the CIA machinery thus limiting the recruitment of the Rli1 client (*Figure 8*). In summary, our mutational studies strongly suggest a model for Yae1-Lto1 acting as a dedicated adaptor complex mediating the contacts between the CIA targeting complex and Rli1. The CIA targeting complex binds Lto1 via its C-terminal W, the two deca-GX$_3$ motifs are crucial for Yae1-Lto1 complex formation, and, as shown earlier (*Zhai et al., 2014*), Yae1 mediates the contact to Rli1 (*Figure 8*). This chain of binding events facilitates efficient Rli1 maturation. Interestingly, a characteristically different but related maturation strategy is followed by the radical SAM Fe-S protein viperin, an interferon-induced antiviral defense component (*Upadhyay et al., 2014*). This protein uses its conserved C-terminal tryptophan to directly associate with the CIA targeting complex. Removal of this residue abolishes complex formation with the CIA targeting complex, assembly of viperin's Fe-S cluster, and consequently antiviral function. The presence of a tryptophan at the viperin C-terminus thus renders an adaptor function like that of Yae1-Lto1 dispensable.

Lto1 has recently been characterized as a protein involved in the biogenesis of the large ribosomal subunit and in the initiation of translation (*Zhai et al., 2014*). Based on our current results this suggested role of Lto1 in protein synthesis seems to be indirect, and rather is mediated via its (primary) function in Rli1 maturation. As noted above, this Fe-S protein performs key functions in ribosomal subunit biogenesis as well as translation initiation, termination, and recycling. For all these functions, its two Fe-S clusters are essential (*Kispal et al., 2005*; *Khoshnevis et al., 2010*; *Barthelme et al., 2011*; *Becker et al., 2012*). Consistent with the (indirect) roles of Lto1 and also Yae1 in ribosomal protein translation, we found protein synthesis defects in cells deficient in these proteins, similar to what we previously described for a deficiency in Rli1 ([*Kispal et al., 2005*] and this study).

The Fe-S clusters of Rli1 were reported to be particularly sensitive to added oxidants suggesting that the clusters may need stabilization, in particular under oxidative conditions (*Alhebshi et al., 2012*). We designed an experimental approach that allowed us to test the potential stabilizing function of Yae1-Lto1 for the Rli1 Fe-S clusters. This involved the combination of blue light-induced rapid degradation of Yae1 and the estimation of the Fe-S cluster stability of radiolabeled Rli1. Depletion of Yae1 by blue light exposure diminished de novo Fe-S cluster assembly of Rli1, yet did not affect the stability of the clusters, clearly documenting that Yae1 (and presumably Lto1) function as maturation rather than stabilization factors for Rli1. This maturation function is particularly needed under oxidative conditions to generate enough functional Rli1 to support normal growth. In contrast, under anaerobic conditions both *YAE1* and *LTO1* are dispensable for normal growth ([*Snoek and Steensma, 2006*]; this work), indicating that their function can be bypassed to some extent in the absence of oxidative stress. Nevertheless, as documented by our radiolabeling assay these proteins are still beneficial for Rli1 maturation under anaerobic conditions. The decreased need to (re-)generate damaged Fe-S clusters of Rli1 may render them dispensable. These findings are fully consistent with a recent study concluding

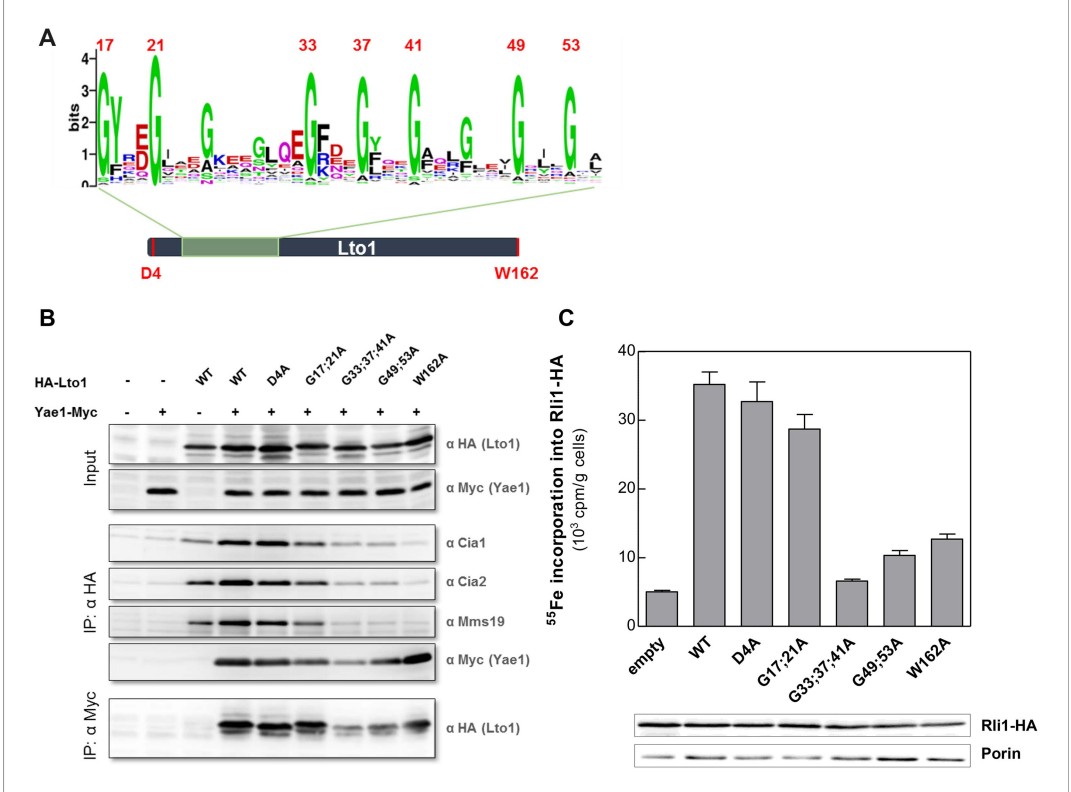

**Figure 6**. The conserved deca-GX$_3$ motif and the C-terminal tryptophan are functionally crucial elements of Lto1. (**A**) Cartoon of *S. cerevisiae* Lto1 to highlight the mutated residues (red) within the deca-GX$_3$ motif and the N- and C-termini (cf. *Figure 2—figure supplement 1B*). (**B**) Gal-NBP35 yeast cells were co-transformed with 2μ vectors containing either no insert or genes encoding HA-Lto1, Yae1-Myc or HA-tagged Lto1 mutants as indicated. Cells were cultivated in minimal medium containing glucose. Cell lysates were prepared, and a trichloroacetic acid (TCA) precipitation was performed to assess the expression levels of the fusion proteins (Input). After HA- or Myc-tag IP the precipitate was analyzed for the indicated tags or proteins by immunoblotting. (**C**) Cells were co-transformed with plasmids containing *RLI1-HA* and single copy plasmids containing *LTO1* or mutated versions as indicated. Cells were cultivated for 24 hr in glucose-containing minimal medium and the $^{55}$Fe radiolabeling-IP procedure was performed to estimate the amount of $^{55}$Fe associated with Rli1-HA. Error bars indicate the SEM (n > 4). The inset shows a representative immunostain of Rli1-HA and porin.

that the oxygen sensitivity of Rli1 is best explained by affecting Fe-S cluster synthesis and/or transfer steps rather than the Rli1 Fe-S cluster stability (*Alhebshi et al., 2012*). This view nicely fits to the observation that *RLI1* functions as a high copy suppressor of *LTO1* mutations (*Zhai et al., 2014*). Collectively, the sum of these studies suggest that Yae1-Lto1 largely increase the efficiency of Rli1 maturation, in particular under aerobic conditions, yet in the absence of oxygen these factors can be bypassed to some extent without losing cell viability.

The sensitivity of Fe-S protein biogenesis pathways towards oxygen and/or oxidative stress has also been noted for members of the bacterial ISC assembly system. For instance, *Azotobacter vinelandii* IscA was required for the maturation of [4Fe-4S] proteins only under high oxygen concentrations (*Johnson et al., 2006*), and *Escherichia coli* IscA-SufA paralogs are needed for Fe-S protein assembly especially under aerobic growth conditions (*Tan et al., 2009*). Similarly, *E. coli* ErpA is essential for Fe-S cluster maturation of isoprenoid biosynthesis enzymes under aerobic conditions, but can be replaced by IscA under anaerobiosis (*Vinella et al., 2009*). As a second example, the bacterial NfuA protein functions as a Fe-S cluster transfer protein that is essential for maturation of dedicated Fe-S proteins such as aconitase particularly in the presence of oxygen or oxidative stress conditions, yet can be bypassed under low oxygen conditions (*Angelini et al., 2008*; *Bandyopadhyay et al., 2008*; *Py et al., 2012*). In comparison and on the contrary, the mitochondrial ISC assembly

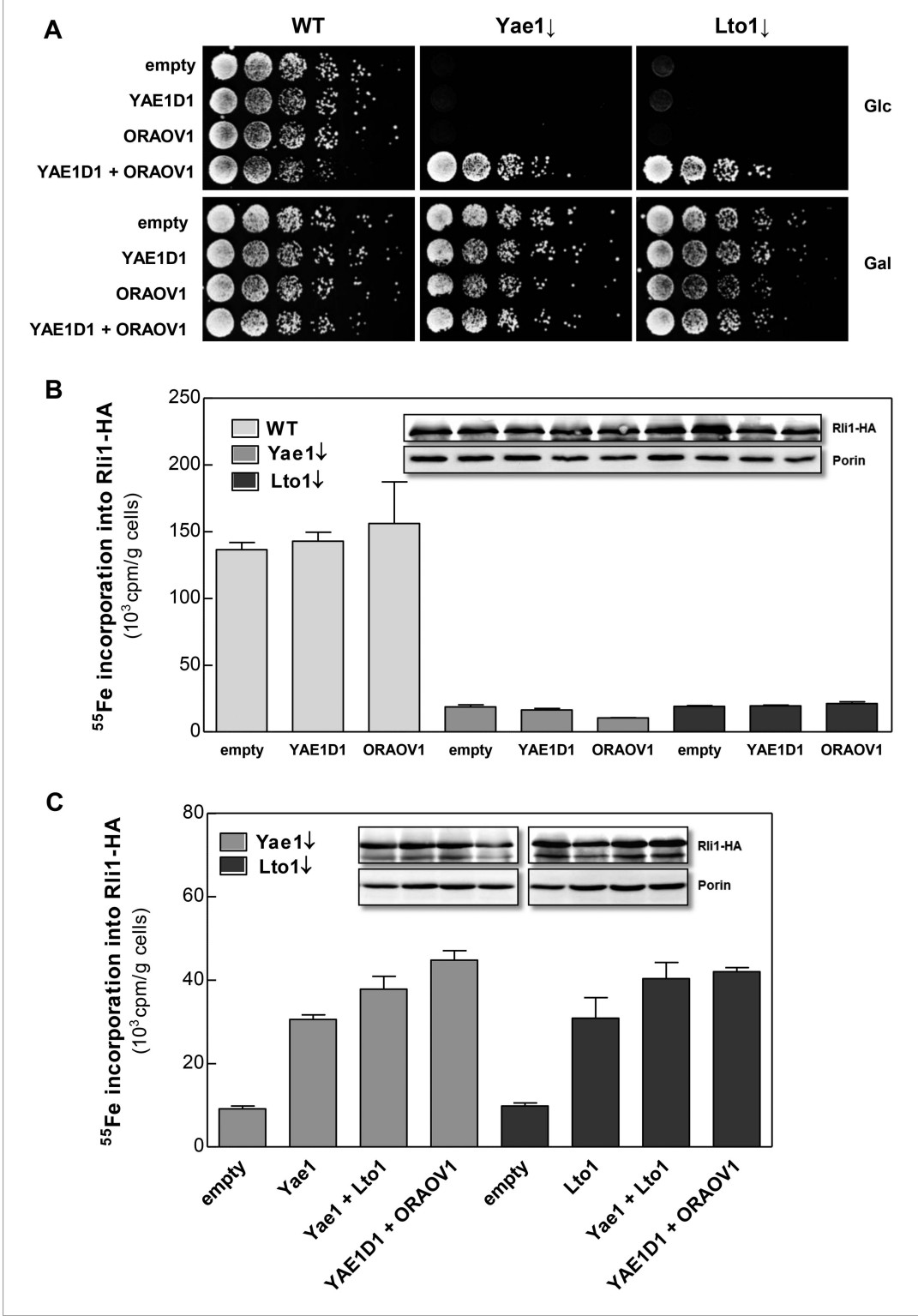

**Figure 7**. The human YAE1D1-ORAOV1 complex can functionally replace Yae1-Lto1. (**A**) WT, Gal-YAE1 and Gal-LTO1 cells were transformed with 2μ vectors containing either no insert (empty) or genes encoding human YAE1D1 and/or ORAOV1 as indicated. Cells were cultivated overnight in minimal medium containing glucose to deplete Yae1 and Lto1 (↓). Serial dilutions (1:5) were spotted onto minimal medium agar plates supplemented with glucose (Glc) or galactose (Gal). (**B**) Cells expressing *RLI1-HA* plus *YAE1D1* or *ORAOV1* were radiolabeled with $^{55}$Fe, and the amount of $^{55}$Fe associated with Rli1-HA was quantified by scintillation counting. Error bars indicate

*Figure 7. continued on next page*

*Figure 7. Continued*

the SEM (n > 4). The inset shows a representative immunostain of Rli1-HA and porin. (**C**) In a similar analysis as in part **B** the effects of the expression of *YAE1*, *LTO1*, *YAE1D1* or *ORAOV1* for Rli1-HA maturation in Yae1- and Lto1-depleted cells were estimated.

pathway is crucial under both aerobic and anaerobic conditions. Deletion of *S. cerevisiae GRX5* encoding the mitochondrial monothiol glutaredoxin largely impairs growth rates on minimal medium even under anaerobiosis, and functional inactivation of the Isa1-Isa2 proteins involved in [4Fe-4S] protein biogenesis does not allow cell growth and mitochondrial [4Fe-4S] protein maturation in the presence or absence of oxygen (*Rodriguez-Manzaneque et al., 2002*; *Muhlenhoff et al., 2011*).

The characterization of Yae1-Lto1 as a dedicated adaptor complex of the CIA machinery for recruitment of specific Fe-S proteins such as Rli1 may be a paradigm for other Fe-S clients and may

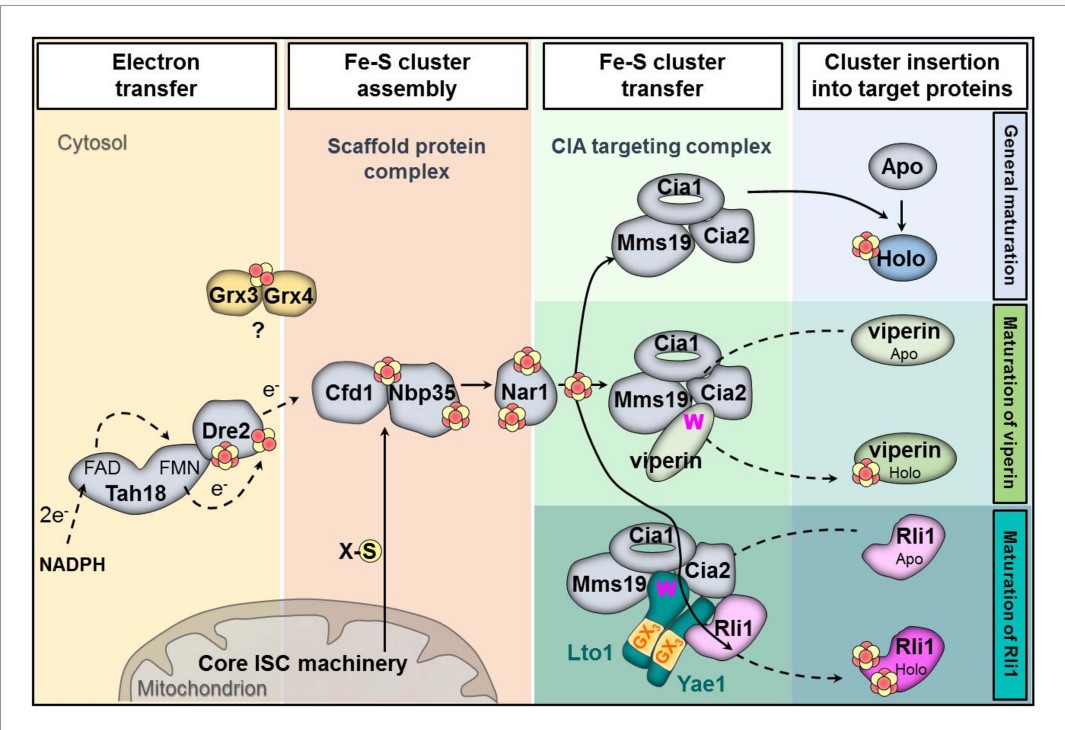

**Figure 8**. Working model for the specific function of the CIA proteins Yae1 and Lto1 in the maturation of the cytosolic Fe-S protein Rli1. Maturation of cytosolic and nuclear Fe-S proteins is a multi-step process conducted by different CIA protein subcomplexes. First, a [4Fe-4S] cluster is assembled on the scaffold protein complex composed of Cfd1-Nbp35. This reaction depends on a yet unknown sulfur-containing compound (X-S) which is produced by the mitochondrial iron-sulfur cluster (ISC) assembly machinery and is exported by the mitochondrial ABC transporter Atm1 to the cytosol. Further, the electron transfer chain NADPH-Tah18-Dre2 is required. The Grx3-Grx4 complex mediates a yet undefined function, possibly in delivering iron, for cytosolic-nuclear Fe-S protein biogenesis. Second, the newly assembled [4Fe-4S] cluster is released from Cfd1-Nbp35 and inserted into target apoproteins via Nar1 and the CIA targeting complex (Cia1-Cia2-Mms19). The precise mechanism of the latter steps in still unclear. As reported in this study, maturation of the essential Fe-S protein Rli1 additionally depends on the function of the two specific adaptor proteins Yae1 and Lto1. The Yae1-Lto1 complex uses a unique binding cascade to recruit Rli1 to the CIA targeting complex for Fe-S cluster insertion. The CIA targeting complex interacts with the conserved C-terminal tryptophan residue (W) of Lto1; the conserved deca-$GX_3$ motifs (yellow boxes) of Yae1-Lto1 are crucial for their complex formation; and Yae1 associates with Rli1. Such an adaptor function is dispensable in the maturation pathway of the virus-induced Fe-S protein viperin because it directly binds to the CIA targeting complex via its conserved C-terminal tryptophan residue.

suggest the existence of additional devoted CIA maturation factors. In mitochondria, several specific ISC assembly factors have been described including the P-loop NTPase IND1 involved in complex I maturation, NFU1 and BOLA3 required for assembly of complex Fe-S proteins such as respiratory complexes I and II and lipoic acid synthase (*Stehling and Lill, 2013*). Their molecular mode of action in the assembly process is still unclear, but an adaptor role similar to Yae1-Lto1 can now be tested. Future elucidation of the molecular mechanisms of these specificity factors will clarify whether they follow common or distinct strategies for Fe-S cluster insertion. Solving their 3D structure will certainly be helpful for understanding how these proteins perform their adaptor role between the late parts of the ISC or CIA machineries and their dedicated Fe-S clients.

## Materials and methods

### Yeast strains and growth conditions

*S. cerevisiae* strain W303-1A was used as WT strain. Detailed information on the yeast strains, oligonucleotides, and expression vectors used in this study is provided in *Supplementary file 1*. Human *YAE1D1* and *ORAOV1* genes (IMAGE ID 4544931/AU101 H10 M13F) were obtained from Eurofins and SourceBioscience, respectively.

Yeast strains were cultivated in rich (YP) or minimal (SC) medium containing the required carbon sources at a concentration of 2% (wt/vol) (*Sherman, 2002*). For growth under anaerobic conditions media were supplemented with 0.2% (vol/vol) ethanol, 0.2% (vol/vol) Tween-80 and 30 µg/ml ergosterol. Low fluorescence medium was used for cultivation of cells expressing psd-fusion proteins (*Taxis et al., 2006*).

### Sample preparation and analysis by LC-MS/MS

TAP was performed as described (*Rigaut et al., 1999*; *Puig et al., 2001*; *Gavin et al., 2002*). Eluted proteins were reduced by 5 mM DTT for 30 min at 56°C, alkylated in 7 mM iodoacetamide for 30 min at room temperature in the dark, and then precipitated in 20% ice-cold trichloroacetic acid. Protein pellets were resuspended in trypsin solution (30 µl of 10% acetonitrile (ACN), 50 mM ammonium bicarbonate (Ambic), and 0.025 µg trypsin) and digested at 37°C overnight. Digestion was stopped by addition of 30 µl of 50% ACN in 5% formic acid (FA). Peptide solutions were dried in a speed vac and resuspended in 0.1% trifluoroacetic acid for StageTIP purification (*Rappsilber et al., 2007*). StageTIPs were prepared with 2 layers of 3 M Empore C18 membranes. Peptides were eluted with 40 µl of 50% ACN in 1% acetic acid and dried in a speed vac prior to MS analysis. After resuspension in 20 µl of 5% ACN in 5% FA, 10% of each sample were analyzed by LC-MS/MS during a 45 min gradient by reversed-phase chromatography coupled to an LTQ-Orbitrap Velos mass spectrometer (Thermo Scientific, Waltham, MA) in Higher Energy Collision Dissociation mode. Peak lists were extracted using MaxQuant (version 1.1.1.36) (*Cox and Mann, 2008*) and submitted to Mascot 2.2.03 (www.matrixscience.com) searches against the yeast SGD protein database including common contaminants, with carbamidomethylation (C) as a fixed modification, oxidation (M), as well as deamidation (N, Q) and protein N-terminal acetylation as variable modifications. Mass error tolerances were set to 10 ppm (MS) and 0.05 Da (MS/MS). Search results were loaded into Scaffold (version 3.3.1, www.proteomesoftware.com) and into ProHITS (*Liu et al., 2010*) for subsequent SAINT analysis (*Choi et al., 2011*).

### SAINT analysis

To extract bait-prey interactions from our TAP-MS data, we loaded our Mascot search results without any Mascot score threshold into the ProHITS database (*Liu et al., 2010*) and made use of the integrated SAINT algorithm (*Choi et al., 2011*) to perform a probabilistic scoring. This algorithm assigns a confidence score to the probability of observing a true interaction using Poisson distributions. In order to assess the dependency of the observed interactions to changes in the SAINT parameters, we applied the SAINT algorithm by using a range of different parameters, for example, the average confidence score threshold (0.8–0.95 in steps of 0.01, range taken from *Choi et al. (2011)*), Mascot scores (10–50 in steps of 10), with or without taking extremely high counts into account, and with or without normalization for spectral counts. In total, we had 320 parameter combinations. By varying these parameters we then counted the number of times that we observed an

interaction against the number of variations, thus giving a fraction between 0 and 1 for each bait-prey pair. We only considered interactions that have been assigned an average SAINT score above 0.8 in at least one of the parameter sets. We then built a network with interactions of high (≥0.5) and low (<0.5) robustness, that is, resistance or sensitivity to variations in SAINT parameters. To support our findings and to expand our network of observed bait-prey interactions, we integrated data from the STRING database (*Szklarczyk et al., 2011*). For interactions observed in our screen, we kept all experimental STRING data without a confidence score threshold. For additional STRING interactions, we only considered experimentally observed interactions with the highest predefined STRING confidence score (0.9).

## Protein–protein interaction in *S. cerevisiae*

For (co-) IP cells were cultivated for 40 hr in glucose-containing minimal medium and 0.5 g cells were resuspended in 0.55 ml lysis buffer (50 mM Tris-HCl pH 7.5, 5% [vol/vol] glycerol, 100 mM NaCl, 1.5 mM MgCl₂, 0.2% [vol/vol] NP-40, 1 mM DTT, 1 mM PMSF) with Complete Protease Inhibitor Cocktail (Roche). Cells were disrupted by vortexing with glass beads (three 1 min bursts) and debris removed (1500×g, 5 min, 4°C). The supernatant was further clarified by centrifugation (13,000×g, 10 min, 4°C). Anti-HA or anti-Myc beads were added to the clarified cell extracts, and the samples rotated for 1.5 hr at 4°C. Beads were washed three times with 0.5 ml lysis buffer and analyzed by immunoblotting using monoclonal antibodies against the HA or Myc-tag (Santa Cruz Biotechnology) or rabbit antibodies raised against the indicated proteins.

## Identification of the physiological translation start site within the *LTO1* gene

During the construction of the *GALL* promoter-containing Yae1 and Lto1 depletion strains we noted that a promoter insertion according to the translation start site of *LTO1* (as listed by the SGD data base) resulted in a strain (termed Gal-LTO1^long) that showed no growth defect under depletion conditions, that is, growth on glucose-containing media (*Figure 2—figure supplement 2B*). This is in contrast to the findings for the Gal-YAE1 strain, and the fact that both genes are essential for cell viability. Upon examination of the reason for the lack of a growth defect, we suspected that the Gal-LTO1^long strain did not allow a critical reduction of the Lto1 protein. A combination of bioinformatic, cell biological and biochemical analyses provided clear information that the reason for this observation was an erroneous translation start site for *LTO1*. As summarized in detail below the correct physiological start site of the *LTO1* gene is located 108 bp downstream of the annotated start codon (ATG3 in *Figure 2—figure supplement 2A*).

The first hint for an annotation error of the *LTO1* gene was provided by a multi-sequence alignment of Lto1 homologs. *S. cerevisiae* Lto1 contained an unusual N-terminal extension of 36 amino acid residues (highlighted in cyan in *Figure 2—figure supplement 1A*) that was not present in any other Lto1 homolog. Inspection of the *S. cerevisiae* Lto1 protein sequence suggested another putative translation start site for Lto1 (residue Met37 corresponding to codon ATG3; *Figure 2—figure supplement 2A*) which roughly matches the N-termini of the other Lto1 homologs. Ribosome foot-printing data (*Ingolia et al., 2009*) supported that the actively translated region of the *LTO1* mRNA does not encompass the N-terminal extension of 36 residues (*Figure 2—figure supplement 3A*). To directly test the importance of the N-terminal 36 residues, we created a *LTO1* mutant in which the *GALL* promoter was inserted right in front of ATG3. This mutant strain (termed Gal-LTO1) showed normal growth on galactose-containing medium demonstrating that the N-terminal 36 residue extension of Lto1 is not critical for cell viability (*Figure 2—figure supplement 2B*). Conspicuously, the Gal-LTO1 cells were unable to grow on glucose-containing medium suggesting efficient depletion of Lto1, unlike that seen for Gal-LTO1^long cells.

To investigate the role of the N-terminal Lto1 segment, we exchanged the coding information for methionine 1 (ATG1) and an additional methionine (residue 10, ATG2) within the 36 residue segment for a stop codon (*Figure 2—figure supplement 3B*). These proteins and the non-mutated Lto1 were expressed from a centromeric plasmid under control of the endogenous promoter in Gal-LTO1 cells grown with galactose or glucose. Both *LTO1* mutations did not affect cell growth. This result clearly documented that the N-terminal segment of Lto1 including its previously proposed start methionine is not essential suggesting that ATG3 rather than ATG1 is the physiologically correct translation start site (*Figure 2—figure supplement 3B*). To further support this conclusion, we analyzed the

importance of the DNA region between codon ATG1 and ATG3 for *LTO1* expression efficiency. To this end, we fused different DNA segments of the *LTO1* gene in front of the luciferase reporter gene (*Figure 2—figure supplement 3C*). The fusion constructs were transformed in WT yeast cells and the luciferase-based luminescence was recorded as a measure of *LTO1* promoter efficiency. Any truncation within the ATG1-ATG3 segment strongly impaired the transcriptional activity suggesting that this region has gene regulatory rather than coding function. Collectively, these findings demonstrate that the physiological start site of *LTO1* is located 108 bp downstream of the previously annotated start codon.

## Miscellaneous methods

The following published methods were used: manipulation of DNA and PCR (*Sambrook and Russell, 2001*), transformation of yeast cells (*Gietz and Woods, 2002*), immunostaining (*Harlow and Lane, 1998*), in vivo labeling of yeast cells with $^{55}FeCl_3$ and measurement of $^{55}Fe$ incorporation into Fe-S proteins by IP and scintillation counting (*Pierik et al., 2009*), pulse labeling of yeast cells with $^{35}S$-methionine (*Kispal et al., 2005*), TAP tag affinity purification (*Puig et al., 2001*; *Gavin et al., 2002*), determination of promoter strength using luciferase (*Molik et al., 2007*), light-induced degradation of psd-fused target proteins (*Renicke et al., 2013*) and enzyme activity measurements (*Molik et al., 2007*). Sulfite reductase activity was assessed in vivo (*Stehling et al., 2012*). In this case, cells were cultivated on rich medium agar plates supplemented with galactose before they were spotted onto minimal medium agar plates supplemented with galactose or glucose and 1% (wt/vol) β-alanin, 0.1% (wt/vol) ammonium bismuth citrate, 0.3% (wt/vol) sodium sulfite.

## Acknowledgements

We gratefully acknowledge the expert technical assistance of Nils Herlerth. RL was supported by Deutsche Forschungsgemeinschaft (SFB 593, GRK1216, SPP 1710), LOEWE program of the state of Hesse, and the Max–Planck Gesellschaft. CT was supported by Deutsche Forschungsgemeinschaft (TA320-3-1, GRK1216), A-CG acknowledges support by the Bundesministerium für Bildung und Forschung (BMBF, 01GS0865).

## Additional information

### Funding

| Funder | Grant reference | Author |
| --- | --- | --- |
| Deutsche Forschungsgemeinschaft (DFG) | SFB 593, GRK1216, SPP 1710 | Roland Lill |
| Max-Planck-Gesellschaft | Fellow | Roland Lill |
| LOEWE Zentrum Synmikro Marburg | | Roland Lill |
| Bundesministerium für Bildung und Forschung | 01GS0865 | Anne-Claude Gavin |
| Deutsche Forschungsgemeinschaft (DFG) | GRK1216, TA320-3-1 | Christof Taxis |

The funders had no role in study design, data collection and interpretation, or the decision to submit the work for publication.

### Author contributions

VDP, Conception and design, Acquisition of data, Analysis and interpretation of data, Drafting or revising the article; UM, A-CG, AJP, RL, Conception and design, Analysis and interpretation of data, Drafting or revising the article; MS, Acquisition of data; JS, KGK, Acquisition of data, Drafting or revising the article; CR, CT, Drafting or revising the article, Contributed unpublished essential data or reagents

### Author ORCIDs

Karl G Kugler, [ID] http://orcid.org/0000-0003-2342-7472
Roland Lill, [ID] http://orcid.org/0000-0002-8345-6518

## Additional files

**Supplementary file**

• Supplementary file 1. *S. cerevisiae* strains, Oligonucleotides and Expression vectors.

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
