## [Decision Letter]

Thank you for submitting your work entitled “The deca-GX3 proteins Yae1-Lto1 function as adaptors recruiting the ABC protein Rli1 for iron-sulfur cluster insertion” for peer review at *eLife*. Your submission has been favorably evaluated by Michael Marletta (Senior editor) and three reviewers, one of whom, Dennis R Dean, served as a guest Reviewing editor.

The reviewers have discussed the reviews with one another and the Reviewing editor has drafted this decision to help you prepare a revised submission.

Summary:

Iron-sulfur clusters are simple inorganic structures that are essential prosthetic groups contained in a variety of proteins. The maturation of proteins that contain Fe-S clusters is complicated, probably because free forms of both iron and sulfur are toxic, and Fe-S clusters contained in many proteins are highly sensitive to oxygen. Fundamental aspects of Fe-S protein maturation involve the separate formation of Fe-S clusters on an assembly scaffold and the subsequent delivery of clusters to a variety of “client” proteins. The manuscript by Paul et al. describes the functional analysis of two cytosolic iron sulfur cluster assembly (CIA) factors, Lto1 and Yae1 that are involved in the specialized delivery of Fe-S clusters to a specific client. Namely, these proteins are shown to be dedicated specificity factors for assembly of the Fe-S clusters contained in Rli1, a protein essential for ribosome function in yeast. The authors provide evidence that depletion of either Lto1 or Yae1 causes a specific loss in maturation of Rli1 but not in maturation of other cytosolic or nuclear Fe-S proteins. It is shown that Lto1/Yae1 form a complex that interacts with the CIA targeting complex. It is also demonstrated that Lto1/Yae1 are involved in cluster delivery for maturation of Rli1, rather than stability of the nascent clusters post maturation. The unique (for eukaryotes) deca-GX3 motif in both Lto1 and Yae1 is shown to facilitate their interaction while a conserved tryptophan in Yae1 is needed for recruitment of the target Rli1. Evolutionary conservation of the function of Yae1 and Lto1 is shown via the demonstration that their function in yeast can be replaced by the human orthologues YAED1 and ORAOV1. It is not yet fully understood why there is a dedicated system for the terminal stage in Rli1 maturation, but this could be related to a particular oxygen sensitivity of its associated clusters.

Essential revisions:

1) In the case of certain bacteria, at least, loss of [Fe-S] cluster transfer proteins (for example, NfuA and IscA) exhibit an inability to grow under conditions of excess oxygen but have no growth phenotype when grown under low oxygen tension. This aspect should be discussed.

2) Some of the Discussion is not necessary as various aspects that are presented in the Discussion recapitulates what has already been presented in experimental rationale and interpretation of the key findings has already been clearly articulated in the Results section. A somewhat abbreviated Discussion that only presents the main conclusion, or perhaps, better, the implications of the work would be sufficient.

3) The investigators make use of a C25S Rli1 mutant to address the oxygen-dependency of Rli1. This experiment was reported by Zhai et al., in 2013 and should be cited.

Minor points:

1) In the fourth paragraph of the Introduction the authors refer to Yae1 and Lto1 as “new” binding partners. They might be newly discovered but they are by no means new.

2) The authors consistently refer to “mutant” proteins when they are actually characterizing proteins that have amino acid substitutions, not mutations.

3) The role of Yae1 and Lto1 in iron-sulfur maturation of Rli1 is shown to be partially by-passed in anaerobic cells. The efficiency of the GAL shutdown in the anaerobic cells is not shown, so it is not clear whether a bypass exists or Rli1 maturation occurs via residual Yae1 and Lto1. Is glucose repression of YAE1 and LTO1 blunted in anaerobic cells?

4) In the Yae1-psd study shown in Figure 4—figure supplement 1 panel B, the experimental details are not easy to understand. Clearly, the depletion of Yae1 impairs 55Fe incorporation into Rli1, but this does not correlate with growth impairment. Obviously, growth impairment will only occur with turnover of holo-Rli1 in the Yae1 attenuated cells.

Enhanced binding of Lto1 and Yae1 with CIA components is shown with depletion of Nbp35. Does depletion of Cia1 or Cia2 result in enhanced binding of Lto1/Yae1 with Rli1?

5) In the first paragraph of the subsection “Yae1 and Lto1 specifically mature the Fe-S protein Rli1”: For the less informed reader it would be helpful to say “canonical nuclear Fe-S target proteins...”

6) In the last paragraph of the subsection “Why are YAE1 and LTO1 non-essential under anaerobic conditions?”. The Cys to Ser mutations in Rli1 and the impact of these mutants in relation to oxygen level is interesting. The authors comment that loss of one ligand in other Fe-S proteins still can allow cluster assembly, true but rather speculative here it seems.

7) In the last sentence of the subsection “Why are YAE1 and LTO1 non-essential under anaerobic conditions?” Perhaps it is better to say the requirement for Yae1 and Lto1 may be lower in anaerobic environments – not the function.

---

## [Author Response]

*1) In the case of certain bacteria, at least, loss of [Fe-S] cluster transfer proteins (for example, NfuA and IscA) exhibit an inability to grow under conditions of excess oxygen but have no growth phenotype when grown under low oxygen tension. This aspect should be discussed*.

This is a great idea. We have added a paragraph to Discussion in which we compare the oxygen sensitivity of Yae1-Lto1-depleted cells with previous findings for the growth and functional behavior of various bacteria deficient in NfuA or A-type ISC proteins in the presence and absence of oxygen. We then compare these results with those for depletion of the *S. cerevisiae* Isa1/2 and Grx5 proteins since in these cases no such behavior was described. Rather, these proteins are needed under both anaerobic and aerobic conditions.

*2) Some of the Discussion is not necessary as various aspects that are presented in the Discussion recapitulates what has already been presented in experimental rationale and interpretation of the key findings has already been clearly articulated in the Results section. A somewhat abbreviated Discussion that only presents the main conclusion, or perhaps, better, the implications of the work would be sufficient*.

We have screened our Discussion for unnecessary repetitions, and tried to shorten the text wherever possible without losing clarity. In general, we think that some repetition of (the most important) findings is needed to also allow readers that do not entirely screen the Results section to comprehend our findings and conclusions. We hope the Reviewers can follow our view.

*3) The investigators make use of a C25S Rli1 mutant to address the oxygen-dependency of Rli1. This experiment was reported by Zhai et al. in 2013 and should be cited*.

We had cited this paper in the beginning of the relevant paragraph. This paper drew a misleading conclusion from the analysis of just this one Rli1 mutant protein, namely that the Rli1 Fe-S clusters are no longer essential under anaerobic conditions. We wanted to avoid this criticism of the (otherwise very nice) Zhai paper. We now have cited this paper again as suggested.

*Minor points*:

*1) In the fourth paragraph of the Introduction the authors refer to Yae1 and Lto1 as “new” binding partners. They might be newly discovered but they are by no means new*.

We have deleted the term “new” in the Introduction.

*2) The authors consistently refer to “mutant” proteins when they are actually characterizing proteins that have amino acid substitutions, not mutations*.

In our understanding (of course, as non-native speakers), the term “mutant protein” does not state that the protein itself was mutated (“mutated protein” would be surely incorrect). We are afraid that the use of other terms such as “altered protein” or “protein carrying amino acid substitutions” might be more confusing to readers than the term “mutant protein” which is generally used by many other authors. We are certainly open to suggestions for better diction.

3) The role of Yae1 and Lto1 in iron-sulfur maturation of Rli1 is shown to be partially by-passed in anaerobic cells. The efficiency of the GAL shutdown in the anaerobic cells is not shown, so it is not clear whether a bypass exists or Rli1 maturation occurs via residual Yae1 and Lto1. Is glucose repression of YAE1 and LTO1 blunted in anaerobic cells?

The GAL promoter shutdown under anaerobic conditions is effective as shown by immunoblotting (see Figure 9), and, on the functional side, by the substantial decrease of ^55^Fe bound to Rli1 in Yae1- and Lto1- (or Nbp35)-depleted cells (Figure 5). The analysis of this aspect by immunoblotting was restricted to the detection of Yae1 because we did not succeed in creating an efficient Lto1 antibody. This may be due to the relatively low abundance of both Yae1 and Lto1 in yeast cells (cf. Ghaemmaghami et al. (2003) Nature 425, 737). As published previously (by systematic yeast strain analyses described in Steensma 2006, and reproduced by [76]), the *YAE1* and *LTO1* genes can be deleted without the loss of cell viability under anaerobic conditions. This finding is mimicked by using the GAL promoter strains, and clearly indicates that a low efficiency bypass of the newly identified adapter function of Yae1-Lto1 must exist (even under gene deletion situations), yet only under anaerobic conditions where the maturation process is not negatively influenced by oxygen toxicity.

Author response image 1.Yae1 depletion is efficient under anaerobic conditions. Wild-type (WT), Gal-YAE1, Gal-LTO1 and Gal-NBP35 cells were transformed with plasmids encoding *RLI1-HA* (+) or no gene (-). Cells were depleted for the indicated proteins (↓) by growth for 30 h in glucose-containing minimal medium under anaerobic conditions and transferred to minimal medium lacking iron for additional 16 h. After radiolabeling for 2 h with ^55^Fe under anaerobic conditions, cell extracts were prepared and levels of indicated proteins analyzed via immunoblotting. Due to the low abundance of Yae1, only weak signals were obtained with our available antiserum.**DOI:**
http://dx.doi.org/10.7554/eLife.08231.020

*4) In the Yae1-psd study shown in panel B, the experimental details are not easy to understand. Clearly, the depletion of Yae1 impairs 55Fe incorporation into Rli1, but this does not correlate with growth impairment. Obviously, growth impairment will only occur with turnover of holo-Rli1 in the Yae1 attenuated cells*.

We know that the light-induced diminution of Yae1 (and Lto1) is less efficient than the *GAL* promoter-mediated protein depletion. The resulting slightly higher residual amount of protein still allows some growth of depleted cells (as opposed to the *GAL* situation). We therefore have added cycloheximide to block the re-synthesis of Yae1 during the light exposure period. This trick (introduced by [52]) enhanced the defects of protein depletion (compare immunoblots in Figure 4 with ) and ^55^Fe association with Rli1. This issue is now better mentioned in revised text. In the growth plates, such a trick cannot be used, of course. We have tried to improve our text to make this complex issue clear.

*Enhanced binding of Lto1 and Yae1 with CIA components is shown with depletion of Nbp35*. *Does depletion of Cia1 or Cia2 result in enhanced binding of Lto1/Yae1 with Rli1?*

The use of Nbp35 in this experiment was intentional. When we deplete CIA proteins in general, we observe that the amount of immediate partner(s) of the depleted CIA protein is also decreased. This is very much the case for the components of the CIA targeting complex, i.e. Cia1, Cia2, and Mms19 which act as a platform for Fe-S cluster insertion (see Figure 10). Since the entire late-acting CIA targeting complex is falling apart when one constituent is absent, we expected that this instability may render the interpretation of such experiments difficult or impossible. Nevertheless, we now have performed the suggested Cia1-depletion experiment (see Figure 11). The outcome of the experiment (to our surprise) confirmed the results of the Nbp35 depletion. However, the amount of Rli1 co-precipitating with Yae1 and Lto1 was reduced, and the interaction analysis was much less specific. This likely is the consequence of the dissociation of the CIA targeting complex as discussed above.

Author response image 2.Depletion of individual CIA targeting complex components affects the levels of its other members. Gal-NBP35, Gal-NAR1, Gal-CIA1, Gal-CIA2, Gal-MMS19 and wild-type cells (WT) were grown in minimal medium supplemented with glucose for 40 h. Cell extracts were prepared, and proteins were analyzed by Western blotting using specific rabbit polyclonal antibodies. Porin was used as a loading control. The instability of the CIA targeting complex upon depletion of single members of this complex is indicated by the red box.**DOI:**
http://dx.doi.org/10.7554/eLife.08231.021

Author response image 3.Wild-type (WT), Gal-NBP35 and Gal-CIA1 yeast cells were co-transformed with 2µ vectors containing either no insert or genes encoding HA-Yae1, HA-Lto1 and Rli1-Myc as indicated. Cells were cultivated in minimal medium containing glucose leading to Nbp35 and Cia1 depletion (↓) in Gal-NBP35 and Gal-CIA1 cells, respectively. Cell lysates were prepared, and a HA-tag immunoprecipitation (IP) performed. The immunoprecipitate was analyzed for the indicated proteins or tags by immunoblotting. The lower band in HA-Lto1-containing lanes is a HA-Lto1 degradation product.**DOI:**
http://dx.doi.org/10.7554/eLife.08231.022

Based on all this new information, we have performed the requested analysis of the interaction between Yae1-Lto1 and Rli1 under Nbp35 depletion conditions. We have added this immunoprecipitation experiment as new Figure 1. On the one hand, the results confirm co-immunoprecipitation of Yae1-Lto1 with Rli1 as shown earlier by [76] for wild-type cells. On the other hand and a new finding, the efficiency of Yae1- Lto1-Rli1 complex formation is not changed by Nbp35 depletion indicating that inactivation of the Fe/S cluster synthesis activity affects CIA targeting complex formation with Yae1-Lto1 more efficiently than Yae1-Lto1 adapter interaction with Rli1.

*5) In the first paragraph of the subsection “Yae1 and Lto1 specifically mature the Fe-S protein Rli1”*: *For the less informed reader it would be helpful to say “canonical nuclear Fe-S target proteins...”*

We have added the term “nuclear” as suggested.

*6) In the last paragraph of the subsection “Why are YAE1 and LTO1 non-essential under anaerobic conditions?”. The Cys to Ser mutations in Rli1 and the impact of these mutants in relation to oxygen level is interesting. The authors comment that loss of one ligand in other Fe-S proteins still can allow cluster assembly, true but rather speculative here it seems*.

We had already supported this statement by two references from our own work. We now have added another reference (71) to make clear that such a behavior is not rare.

*7) In the last sentence of the subsection “Why are YAE1 and LTO1 non-essential under anaerobic conditions?” Perhaps it is better to say the requirement for Yae1 and Lto1 may be lower in anaerobic environments – not the function*.

We have replaced the term “function” with “requirement” as suggested.